# Metabolomic Analysis on the Mechanism of Nanoselenium Biofortification Improving the *Siraitia grosvenorii* Nutritional and Health Value

**DOI:** 10.3390/foods11193019

**Published:** 2022-09-29

**Authors:** Chunran Zhou, Jingbang Zhang, Yangliu Wu, Haiyan Cheng, Qiuling Pang, Yuanhui Xiao, Dong Li, Canping Pan

**Affiliations:** 1Innovation Center of Pesticide Research, Department of Applied Chemistry, College of Science, China Agricultural University, China Yuanmingyuan West Road 2, Beijing 100193, China; 2Guangxi Academy of Specialty Crops, Putuo Road 40, Guilin 541004, China; 3Key Laboratory of Green Prevention and Control of Tropical Plant Diseases and Pests, Ministry of Education, College of Plant Protection, Hainan University, Haikou 570228, China

**Keywords:** nanoselenium, *Siraitia grosvenorii*, quality, phenylpropane pathway, volatile organic compounds

## Abstract

Nanoselenium (nano-Se) foliar application is crucial for enhancing plant health. However, the mechanism by which nano-Se biofortification promotes the nutritional components of *Siraitia grosvenorii* remains unclear. In this study, nano-Se foliar application increased the carbohydrate and amino acid contents, including glucose (23.6%), fructose (39.7%), sucrose (60.6%), tryptophan (104.5%), glycine (85.9%), tyrosine (78.4%), phenylalanine (60.1%), glutamic acid (63.4%), and proline (52.5%). Nano-Se application enhanced apigenin (3.8 times), syringic acid (0.7 times), and 4-hydroxy-3,5-dimethoxycinnamic acid (1.4 times) of the phenylpropane biosynthesis pathways. Importantly, the SgCDS (31.1%), CYP-P450 (39.1%), and UGT (24.6%) were induced by nano-Se, which enhanced the mogroside V content (16.2%). Compared to the control, nano-Se treatment dramatically enhanced aromatic substances, including 2-butanone (51.9%), methylpropanal (146.3%), n-nonanal dimer (141.7%), pentanal (52.5%), and 2-pentanone (46.0%). In summary, nano-Se improves *S. grosvenorii* quality by increasing nutrients and volatile organic compounds and adjusting the phenylpropane pathway.

## 1. Introduction 

*Siraitia grosvenorii* (known as Luo-Han-Guo or “fairy fruit”) is one plant with a high economic value. *S. grosvenorii* reportedly cures cough, expectoration, sore throat, and cold [1]. In 2017, the U.S. Food and Drug Administration listed the *S. grosvenorii* fruit as a natural sweetener [2], a sweetness which is 250 times higher than sucrose. Qing et al. indicated that the *S. grosvenorii* fruit potentially has immune-regulating, anti-oxidative, anti-obesity, anti-cancer, and antidiabetic effects [3]. Mogroside V is considered the main biological constituent of *S. grosvenorii* sweetness, attributable to the cucurbitane triterpenoid compounds in the *S. grosvenorii* fruit. Many studies confirmed that the main substances of *S. grosvenorii* are polysaccharides, triterpenoids (especially the cucurbitane triterpenoid), flavonoids, and proteins [4]. Consequently, the improvement of the key nutrients of the *S. grosvenorii* plant needs to be deeply studied.

To date, exogenous factors including plant hormones, chemical elements, and nanomaterials are important in plant health [5,6]. Liu et al. showed that 1 mg/kg of salicylic acid (SA) application stimulated cucumber seedling antioxidation by activating the antioxidant system and decreasing the thiamethoxam (THIM), hymexazol (HMI), and chlorantraniliprole (CAP) accumulation [7]. Foliar application of silicon (2 mM) decreased the malonaldehyde (MDA) levels, thereby alleviating the adverse influence of melon (*Cucumis melo* L.) autotoxicity. Silicon significantly increased the soluble sugar, starch, and α-amylase contents of melon [8]. The foliar application of manganese (Mn) induced the resistance against *C. lagenarium* in cucumber by accumulating lignin, callose, and a reduction of pathogen-induced cell death. These findings provide the scientific basis for exogenous substances application to improve plant quality [9]. Selenium (Se), the main active ingredient of glutathione peroxidase, is crucial for plant antioxidant capacity and quality by regulating primary and secondary metabolism, including plant hormone signals and the phenylpropane pathway. Appropriate Se concentrations improve plant resistance by enhancing the flavone and phenolic acid contents [10]. Nano-Se showed higher bioactivity and lower toxicity than selenate and organic Se [11]. Foliar application of nano-Se (20 mg/L) remarkably increased nutrient levels and promoted phenylpropane metabolism and the fatty acids pathway in pepper plant [12]. Sheikhalipour et al. discovered that chitosan-selenium nanoparticle (Cs-Se NP) application increased the quality and yield of *Momordica charantia* by enhancing photosynthetic capacity. The Cs-Se NP also enhanced the contents of phenols, flavonoids, and essential oils to alleviate salt stress in bitter melon fruits [13]. Therefore, nano-Se biofortification shows great potential in plant health improvement. However, there is limited information on how nano-Se improves *S. grosvenorii* metabolism. 

Metabolomics is a recent development applicable in many fields, including plant, animal, medical science, synthetic biology, and medicine [14]. The changes in metabolic pathways are identified by qualitatively and quantitatively analyzing small molecule compounds. In tomato, Li et al. detected 540 annotated metabolites in 20 metabolic regulatory networks, providing a valuable reference for identifying key regulatory metabolites in other crops [15]. Elsewhere, the metabolomics application of *Klebsiella oxytoca* P620 enhanced the activities of antioxidant enzymes and decreased the concentrations of stress-related metabolites under p-hydroxybenzoic acid (PHBA) stress in cucumber [16]. Ultra-high-performance liquid chromatography coupled to triple quadrupole mass spectrometry (UPLC-QQQ-MS )evaluation of Pu-erh taste and quality under different fermentation and storage time also identified differential metabolites of gallic acid, acetylamino acids, purine alkaloids, pyrimidine alkaloids, and non-glycoside flavonoids [17]. Overall, the above cases demonstrate that metabolomics has been widely studied in plant growth, nutritional quality, food processing, and external stress. Ma et al. indicated that the combined application of shellac and tannic acids increased the shelf life and quality by reducing lipid peroxidation and increasing VOCs in mangoes [18]. In melon, high levels of esters, aldehydes, alcohols, ketones, and acids were detected among 39 melon cultivars by chromatography–mass spectrometry (HS-SPME-GC-MS). This work on the aroma and quality of melon cultivars provides a reference for the genetic modification of other plants [19]. However, there is no metabonomic analysis report on *S. grosvenorii* fruit after quality improvement using nano-Se, despite the immense potential of volatile organic compounds (VOCs) in plant quality. 

Therefore, this study aimed at exploring the potential mechanism of nano-Se (5 mg/L) application on the quality of the *S. grosvenorii* plant. Carbohydrates and amino acids were determined in Se-enriched *S. grosvenorii* and controls. Levels of specific mogroside V metabolites and related synthases of SgCDS, CYP-P450, and UGT also were measured. The differential expression of secondary metabolites and representative aroma components were identified to analyze the metabolic pathways and flavor with nano-Se biofortification. In summary, nano-Se application ameliorates the *S. grosvenorii* quality by regulating the nutritional ingredients, volatile compounds, and secondary metabolites. 

## 2. Materials and Methods

### 2.1. Plant Material and Field Experiment

The experiment was performed in Zhongyong village, Lingui District, Guilin, Guangxi Province, China. The experimental field was divided into two plots, each measuring 667 m^2^. The soil properties are red loam with 42.2 g/kg organic matter content, pH 4.9–5.1, 2.2 g/kg total nitrogen, and 25.9 mg/kg effective phosphorus. The nano-Se concentration and dimensions (50–78 nm) were based on a previous study [20]. Nano-Se (5 mg/L) improved plant quality compared to the control. The nano-Se solution (1500 mg/L) was diluted 300 times with purified water and conducted on a foliar application (5 mg/L concentration). 

Throughout the trial, conventional orchard management was conducted per field experimentation. Every month, beginning 15 May 2020 through 1 August 2020, the nano-Se (5 mg/L) was sprayed on *S. grosvenorii* fruits. In total, nano-Se was sprayed four times, and the control group received the same treatment with water. *S. grosvenorii* fruits were collected on 15 September 2021, and the fruits were weighed. At least 50 *S. grosvenorii* fruits were collected from each treatment to guarantee sufficient specimens. The collected fruit samples were washed with water to remove nano-Se residues from the surface before testing. Washed samples were quickly transferred to the laboratory and stored in a refrigerator at −80 °C. 

### 2.2. Mogroside V Contents and Related Key Enzyme Analyses

*S. grosvenorii* fruits were ground in liquid nitrogen, and 50 mg of freeze-dried powder was added to 1 mL of methanol. The mixed solution was shaken for 5 min, treated using ultrasound for 30 min at 30 °C, and centrifuged at 10,000 rpm for 5 min. The supernatant was purified using 50 mg PSA in a 2 mL centrifuge tube and filtered using a 0.22 µm nylon syringe. The UPLC-MS/MS (Agilent G6465B triple quadrupole, Santa Clara, CA, USA) was equipped with an HPLC reverse phase C18 column (Eclipse Plus C18 2.1 × 50 mm, 1.8 μm). The mobile phases, A and B, were methanol and ultrapure water (V:V = 80:20), respectively. The flow rate was 0.4 mL/min with 5 µL injection volume. The MS was performed using multiple reaction monitoring (MRM) and negative electrospray ionization. The special instrument parameters for mogroside V analysis are provided (Appendix A). 

The activities of squalene epoxidase (SE), cucurbitadienol synthase (SgCDS), cytochrome P450 (CYP-P450), UDP-glucosyltransferase (UGT), and epoxide hydrolase (EPH) were measured using the corresponding standard kits from Jiangsu Enzyme Free Industrial Co., Ltd. (Yancheng, China).

### 2.3. Flavonoids and Phenolic Acid Compounds Analyses

*S. grosvenorii* fruits (100 mg) were extracted with 1 mL of 60% ethanol in water [12]. The homogenate was subjected to ultrasound for 30 min and centrifuged at 12,000 rpm for 5 min. All supernatants were collected by repeating the operating method twice. The mixture was dried by blowing with nitrogen and maintained at a constant volume of 100 mL with 60% ethanol. The solutions were purged with 100 mg C18 and filtrated using a 0.22 μm nylon syringe. The Agilent G6465B triple quadrupole UPLC-MS/MS was equipped with an HPLC reverse phase C18 column (Eclipse Plus C18 2.1 × 50 mm, 1.8 μm). The flow rate was 0.4 mL/min and 5 µL injection volume. The mobile phases A and B consisted of acetonitrile and 0.1% formic ultrapure water, respectively. The instrument parameters of flavonoids and phenolic acid compounds are provided in Appendix A.

### 2.4. Carbohydrate Analyses

*S. grosvenorii* fruits (20 mg) were extracted with 1 mL of ultrapure water. The mixed solution was subjected to ultrasound for 30 min and centrifuged for 10 min at 14,000 rpm. All the supernatants were filtered with a 0.1 μm injection syringe. The filtered liquid was diluted 20 times with ultrapure water. The concentrations of carbohydrates were determined using the Thermo Scientific Dionex ICS-5000^+^ ion chromatograph (Thermo Scientific, Waltham, MA, USA), equipped with an SP single pump, EG eluent generator, AS-AP automatic sampler, DC electrochemical detector, and the Chromeleon version 7.2 SR5 chromatographic data analysis software. The flow rate was 1 mL/min and 10 µL injection volume. Mobile phases A and B for gradient elution were ultrapure water and 200 mM NaOH (50% NaOH diluted with Watson’s water). The detailed instrument parameters of carbohydrates are provided in Appendix A.

### 2.5. Amino Acid Analyses

Briefly, 20 mg of freeze-dried fruit powder was added to 1 mL of Watson’s water. The solution was ultrasound treated for 30 min and centrifuged at 10,000 rpm for 10 min at 4 °C. The supernatant was derivatized with 6-aminoquinolyl-N-hydroxy-succinimidyl carbamate (AQC) before the test. The concentration of amino acid was determined using Liquid chromatography–high resolution mass spectrometry (LC-HRMS). The quantitative analysis of amino acids was determined using the Xcalibur software (Thermo Scientific, Waltham, MA, USA). The related instrument parameters are provided in Appendix A.

### 2.6. Volatile Compounds Analyses

The volatile compounds were analyzed using the Gas Chromatography–Ion Mobility Spectrometer (GC-IMS). Approximately 0.5 g of *S. grosvenorii* fruits were placed in a 20 mL headspace vial and incubated at 60 °C for 20 min before determination. The analysis software was set as previously described [18]. Briefly, the volatile compounds were qualitatively analyzed using the NIST and IMS databases. Each sample was analyzed from a different angle using three plugins (Reporter, Gallery Plot, and Dynamic PCA) and VOCAL.

### 2.7. Widely Targeted Metabolomics Analyses

Metabolomic extraction and analysis were conducted by Metware Biotechnology Co., Ltd. (Wuhan, China). Freeze-dried powdered samples (100 mg) were extracted with 1 mL of 70% methanol [21]. The mixture was violently shaken for the 30 s every 30 min, and the procedure was repeated six times. The homogenate was extracted overnight at 4 °C. The extracted homogenate was centrifuged for 10 min at 12,000 rpm and filtrated through a 0.22 μm pore. The collected sample was analyzed using UPLC-MS/MS equipped with a column of Agilent SB-C18 1.8 μm, 2.1 × 100 mm. Acetonitrile (containing 0.1% formic acid) and 0.1% formic ultrapure water were used as mobile phases A and B, respectively. The flow rate was 0.35 mL/min with 4 µL injection volume. The gradient elution was as follows: 0–9 min, 5–95% B; 9–10 min, 95% B; 10–11 min, 95–5% B; and 11–14 min, 5% B. The metabolisms were identified using the Metware database (MWDB) and quantified based on their peak areas.

### 2.8. Statistical Analysis

The graphs were structured with GraphPad Prism Version 8.0 (San Diego, CA, USA). Statistical analysis was performed using SPSS 26.0. Tukey’s test (*p* < 0.05) was used to identify different treatments, and the MetaboAnalyst was performed to analyze the metabolisms.

## 3. Results

### 3.1. The Se Content and S. grosvenorii Fruit Weight

Selenium is a beneficial element in plant growth. Nano-Se foliar application significantly increased the Se level (3.7 times) compared with the control (Appendix A). In addition, nano-Se foliar application enhanced the weight of *S. grosvenorii* fruit by 70.4% (Appendix A).

### 3.2. Mogroside V and the Related key Enzyme Contents in S. grosvenorii

Nano-Se foliar application improved the levels of mogroside V and related key enzymes. Nano-Se (5 mg/L) increased the mogroside V by 16.2% compared to the control (Figure 1). The SgCDS, CYP-P450, and UGT levels increased by 31.1%, 39.1%, and 24.6%, respectively. However, the SE and EPH contents were unaffected.

### 3.3. Flavone and Phenolic Acids Contents on S. grosvenorii

Flavone and phenolic acids are crucial in the improvement of plant antioxidant ability. Nano-Se foliar application increased apigenin levels (3.8 times), but luteolin, quercetin, and kaempferol did not change relative to the control. The contents of chlorogenic acid, caffeic acid, syringic acid, 4-hydroxy-3,5-dimethoxycinnamic acid, ferulic acid, and 4-hydroxybenzoic acid increased by 23.5%, 28.4%, 72.5%,138.0%, 22.9%, and 80.4%, respectively (Figure 2).

### 3.4. The Carbohydrate Contents on S. grosvenorii

Glucose, fructose, sucrose, galactose, mannose, and fucose levels were tested to compare the nano-Se effect on the carbohydrate content of *S. grosvenorii*. Glucose, fructose, and sucrose levels increased by 23.6%, 39.7%, and 60.6%, respectively, in nano-Se-treated plants than the control (Figure 3), while galactose, mannose, and fucose contents had non-significant change.

### 3.5. The Contents of Amino Acids on S. grosvenorii

Amino acids are important compounds in plant health. The PCA showed a distinct separation between the nano-Se-treated and control samples, implying a considerable difference (Figure 4A). Nano-Se treatment increased histidine (108.7%), tryptophan (104.5%), serine (87.4%), glycine (85.9%), lysine (67.5%), tyrosine (78.4%), aspartic acid (65.2%), phenylalanine (60.1%), valine (62.0%), glutamic acid (53.4%), and proline (52.5%) levels (Figure 4C). However, the aminobutyric acid, ammonium chloride, and threonine levels were unchanged across treatments.

### 3.6. Volatile Compounds Analyses on S. grosvenorii

The volatile compounds were determined using GC-IMS, and samples from different treatments were distinctly separated (Figure 5A). There was a remarkable difference between the control and nano-Se treatment. The data are presented in a three-dimensional spectrum (the *x*-axis is the ion migration time, the *y*-axis is the retention time, and the *z*-axis represents the peak height of quantification), which intuitively shows the different levels of volatile compounds across treatments. The background is blue and the reactive ion peak (RIP) and migration were normalized (Figure 5B,C). Each point on the side of the RIP means one volatile compound. The differences comparison mode was adopted to compare sample differences on the platform. The control sample served as the reference and deducted the reference from the spectrum of the treated samples. If the two VOCs are consistent, the background after deduction is white. The red color indicated high concentration, and the blue color indicated lower content. Volatile substances were distinguished based on the color depth (Figure 5D).

Detailed results of the VOCs qualitative analysis are provided (Figure 5E,F). A total of 30 VOCs, including aldehydes (16), ketones (5), esters (5), alcohols (3), and acids (1), were measured in *S. grosvenorii* fruits. The complete information and differences among samples were intuitively obtained by comparing gallery plots of the volatiles. The numbers on the fingerprint indicated the undetermined compounds in the migration spectrum library. The types and concentrations of volatiles in the *S. grosvenorii* fruits were substantially different with different treatments (Appendix A). The retention index, migration time, and peak volume of volatile substances in each sample are shown (Appendix A).

### 3.7. Effects of Nano-Se Application on the Metabolism of S. grosvenorii Fruits

A widely targeted metabolomics technique was used to determine the types and concentrations of metabolites and therefore explore how nano-Se improves the quality of *S. grosvenorii* fruits. A total of 194 metabolites were detected based on the UPLC-MS/MS. Sixty-two compounds significantly changed by ≥2 or ≤0.5-fold change and ≥1 VIP in treated samples compared to the control. The PCA, PLS-DA, and volcano plots are shown in Figure 6A–C. The control and treatment separated, implying that nano-Se application affected plant metabolism. In addition, the Kyoto Encyclopedia of Genes and Genomes (KEGG) analysis indicated similar classifications for substances in the control and nano-Se treatments. These substances are mainly involved in the biosynthesis of phenylpropanoid, flavonoid, flavone, and flavonol (Figure 6D).

The unit variance scaling (UV) method was used to normalize the compounds with marked differences and conveniently display the change of metabolites. The heat map was drawn by R software (Figure 6E). Twenty-six metabolites were remarkably enhanced, and 36 compounds declined relative to the control (Appendix A). The 26 substances included phenolic acids, coumarins, monoterpenoids, flavonoids, chalcones, dihydroflavone, flavonol, alkaloids, lignans, triterpene, and triterpene saponin. Appendix A shows the detailed information on metabolites among different treatments.

## 4. Discussion

*S. grosvenorii* is used as medicine and food in China. The fruit of *S. grosvenorii* is rich in polysaccharides, glucosides, protein, and vitamins. Recent research on *S. grosvenorii* focused on the therapeutic effects of *S. grosvenorii* extracts on various types of inflammation. There is considerable evidence on the use of exogenous substances to improve plant nutrients, antioxidant capacity, and metabolism [6,11]. Nevertheless, the potential mechanism of nano-Se bioaugmentation to enhance the nutritional components of *S. grosvenorii* plants is obscure. In this study, nano-Se application regulated the phenylpropane pathway and enhanced nutrient levels and VOCs, thus enhancing the *S. grosvenorii* quality.

Mogroside V is the main active constituent of *S. grosvenorii*, an important index for evaluating the *S. grosvenorii* quality. The anti-inflammatory effect of mogroside V has currently attracted extensive research. Di et al. showed that mogroside V inhibits inflammation induced by lipopolysaccharides by up-regulating the expression of protective genes and down-regulating the key inflammatory genes. *S. grosvenorii* had anti-inflammatory, antidiabetic, and anticancer effects [22]. Mogroside V extracts are rich in 1–3 glucose residues, identified as low-polar *S. grosvenorii* glycosides (L-SGgly). The L-SGgly is a proven antidiabetic with significant effects. Zhang et al. showed that L-SGgly relieves diabetes by improving the intestinal flora imbalance, increasing the acetic and butyric acid contents, and enhancing insulin resistance to maintain blood glucose levels [23]. In another study, mogroside extract (MGE) showed excellent antiglycative activities by mitigating the glucose-mediated protein glycoxidation and cross-linking in vitro. Therefore, MGE has antioxidant properties and is a potential antiglycation factor for diabetes treatment [24]. Moreover, mogroside treatment protects against bleomycin-induced pulmonary fibrosis by inhibiting oxidative stress, inflammatory infiltration, and collagen disposition [25]. To date, few studies have focused on the relationship between exogenous compounds and the biosynthesis of mogroside V. This study demonstrated that nano-Se (5 mg/L) application enhanced mogroside V and the related synthetase (SgCDS, CYP-P450, and UGT) contents (Figure 1).

Notably, 62 metabolites affected by nano-Se were identified by the target metabolomic method. The heatmap showed that 26 metabolites increased, but 36 metabolites decreased (Figure 6). These observations imply that nano-Se application influenced various metabolic pathways, such as the phenylpropane pathway in *S. grosvenorii* plants. The syringic acid, 4-hydroxy-3,5-dimethoxycinnamic acid, and apigenin increased by 0.7, 1.4, and 3.8 folds (Figure 2), respectively. Plant phenylpropane metabolism shows great potential in plant health and quality. Nano-Se significantly (*p* < 0.05) improved nutrient component levels than sodium selenite by adjusting phenylpropane and capsaicinoid synthetic pathways and the related genes, promoting pepper quality [12]. Low Se concentration promotes plant growth, but high concentrations are inhibitory [20]. In the *Cardamine violifolia* plant, 200 mg/L selenite biofortification promoted plant growth, nutrient levels, and antioxidant capacity [26]. The citrus peel contains abundant polyphenolic compounds (phenolic acids, flavonoids, and coumarins) usable in traditional medicine. These phenolics and flavonoids are the primary antioxidants protecting plants against free-radical damage [27]. Nano-Se application improved the *S. grosvenorii* quality by increasing the flavonoids and phenolics levels.

The interaction of sugars and amino acids affects fruit quality and aroma. In this work, nano-Se application enhanced carbohydrate compounds and amino acids. Glucose, fructose, and sucrose increased by 23.6%, 39.7%, and 60.6% (Figure 3), respectively. Soluble sugars such as sucrose, glucose, and fructose are crucial for regulating plant growth and development and defense against abiotic stress. Presently, numerous measures were adopted to improve plant quality by increasing soluble sugars [28]. Starch decomposition and soluble sugars accumulation determine the enhanced sweetness evident at the late stage of fruit development [29]. Zhao et al. proposed sugars (soluble sugars and sugar alcohols) as the primary energy sources in fruits contributing to plant quality and taste. In their research, SA (1 µmol/L) treatment adjusted sugar metabolism by increasing sucrose and related genes, thus extending cold storage time in peaches [30]. Nineteen of the twenty-three studied amino acids notably increased (Figure 4) under the nano-Se treatment in this study. Free amino acids are significant in protein synthesis, cell signaling, metabolism, physiology, and health [31]. In addition, nano-Se application increased the aromatic amino acids (tryptophan, phenylalanine, and tyrosine). Tryptophan and tyrosine are precursors of tryptophol and tyrosol, respectively, the substances with important antioxidant activity. In addition, phenylalanine synthesizes 2-phenylethanol, the source of pleasant fragrance [32]. Amino acids and secondary metabolites are vital for the aroma, freshness, and mellowness of tea. Guo et al. revealed the mechanism of environment-induced amino acids biosynthesis in Ginkgo leaves. Amino acids commonly exist in Ginkgo leaves and are probably the main contributors to Ginkgo tea flavor and health benefits [33]. Interestingly, Liu et al. showed that Na_2_SeO_3_ (2 mg/mL) foliar application enhances the photosynthesis and antioxidant enzyme levels in tea plants under low temperatures. Concurrently, Se treatment increased sugar accumulation, enhanced phenylalanine, lysine acid, glutamate, and arginine synthesis, and significantly improved tea quality [34]. Nano-Se treatment could improve *S. grosvenorii* quality by enhancing sugar and amino acids contents, consistent with the above reports.

Volatile compounds are effective for fingerprinting and plant quality control. In Wei et al., the comprehensive characteristics and quality standards of five species of Chinese medicine were established according to the information of volatile spectrum [35]. For this study, nano-Se biofortification enhanced VOCs levels in the *S. grosvenorii* fruit. Levels of 1-pentanol dimer, 2-butanone, 2-pentanone, 3-methylbutanal, ethanol, heptanal dimer, methylpropanal, n-nonanal dimer, octanal, pentanal, and acetone contents increased in *S. grosvenorii* fruit with nano-Se treatment. Nevertheless, the (E)-2-octenal dimer, 2-heptenal (E) dimer, benzaldehyde, ethyl acetate, and propionaldehyde levels were decreased with nano-Se treatment (Appendix A). The results of GC-IMS analysis indicated that the variation of 2-butanone, 2-pentanone, 3-methylbutanal, octanal, and pentanal acted as the characteristic compounds for understanding the effects of varieties and cooking on sorghum quality [36]. Elsewhere, Song et al. identified 2-butanone, ethanol, octanal, and pentanal as the characteristic compounds in quinoa [37]. The flavor and aroma of walnut plants are attributed to the abundance of compounds from different chemical classes existing at low concentrations, including aldehydes, alcohols, ketones, and esters. The quality of walnuts was determined by the volatile compounds, including pentanal, hexanal, nonanal, 2-decenal, and hexanol [38]. Previous study revealed that nano-Se ameliorated green tea’s ability to mitigate the effects of pesticides-induced oxidative stress by regulating the aroma levels, phenylpropane metabolism, and antioxidant ability [39]. Additionally, nano-Se (5 mg/L) biofortification enhanced the volatile compounds (1-popanol, isopentyl alcohol, 2-butanone, 2-heptanone, acetoin, acetone, benzaldehyde, and ethyl-2-methylpropanoate), secondary metabolism, and the related genes to improve wheat resistance to aphids [40]. These observations explain why nano-Se application greatly improved the *S. grosvenorii* aroma. The interaction between the volatiles and plant metabolites potentially explains the mechanism by which nano-Se (5 mg/L) improves the *S. grosvenorii* quality.

## 5. Conclusions

In this study, the levels of mogroside V and the key synthetic enzymes increased with nano-Se treatment. Nano-Se biofortification enhanced the sugar and amino acids contents more than the control, thus improving the *S. grosvenorii* nutrition levels. Furthermore, the metabolomics results demonstrated that nano-Se foliar treatment activated phenylpropane biosynthesis pathways. Nano-Se treatment also increased apigenin, apigenin-7-O-(6’’-acetyl) glucoside, syringic acid, and sinapic acid contents. The major finding of this study is that VOCs related to aroma (1-pentanol dimer, 2-butanone, 2-pentanone, 3-methylbutanal, heptanal dimer, methylpropanal, n-nonanal dimer, octanal, and pentanal) were enhanced. Altogether, nano-Se biofortification improved the quality of *S. grosvenorii* by synergistically increasing mogroside V, carbohydrates, amino acids, VOCs, and regulating the phenylpropane pathway.

## Figures and Tables

**Figure 1 foods-11-03019-f001:**
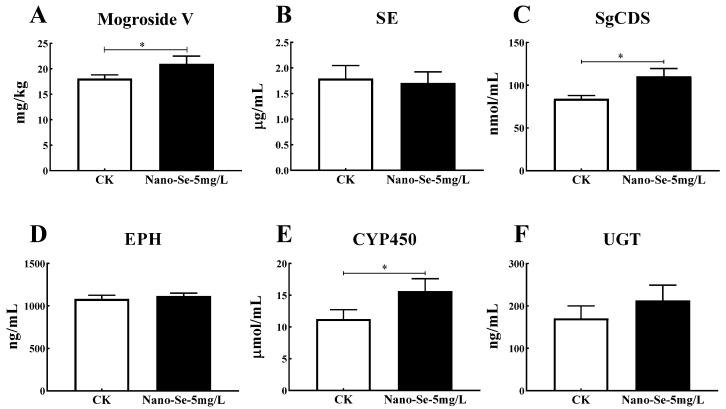
Effects of nano-Se on mogroside V (**A**), SE (**B**), SgCDS (**C**), EPH (**D**), CYP450 (**E**), and UGT (**F**) contents in *S. grosvenorii*. Data are presented as mean ± SEM (* *p* < 0.05. *n* = 3). Squalene epoxidase (SE), cucurbitadienol synthase (SgCDS), epoxide hydrolase (EPH), cytochrome P450 (CYP450), and UDP-glucosyltransferase (UGT).

**Figure 2 foods-11-03019-f002:**
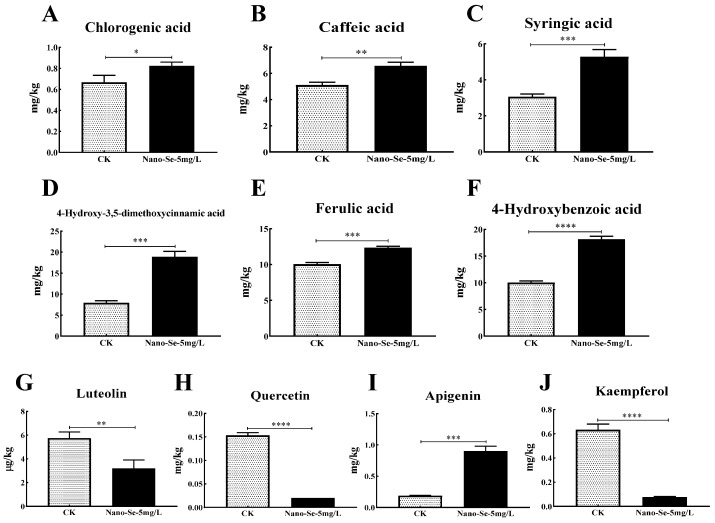
Effects of nano-Se on chlorogenic acid (**A**), caffeic acid (**B**), syringic acid (**C**), 4-hydroxy-3,5-dimethoxycinnamic acid (**D**), ferulic acid (**E**), 4-hydroxybenzoic acid (**F**), luteolin (**G**), quercetin (**H**), apigenin (**I**), and kaempferol (**J**) levels in *S. grosvenorii*. Data are presented as mean ± SEM (* *p* < 0.05, ** *p* < 0.01, *** *p* < 0.001, **** *p* < 0.0001. *n* = 3).

**Figure 3 foods-11-03019-f003:**
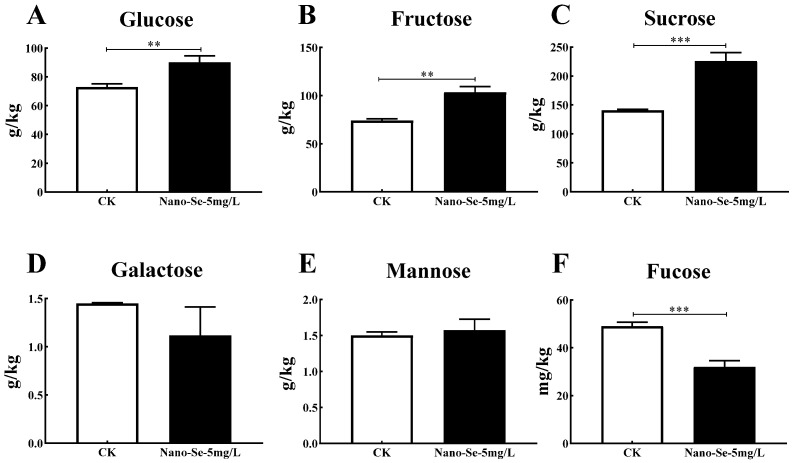
Effects of nano-Se on glucose (**A**), fructose (**B**), sucrose (**C**), galactose (**D**), mannose (**E**), fucose (**F**) contents in *S. grosvenorii*. Data are presented as mean ± SEM (** *p* < 0.01, *** *p* < 0.001. *n* = 3).

**Figure 4 foods-11-03019-f004:**
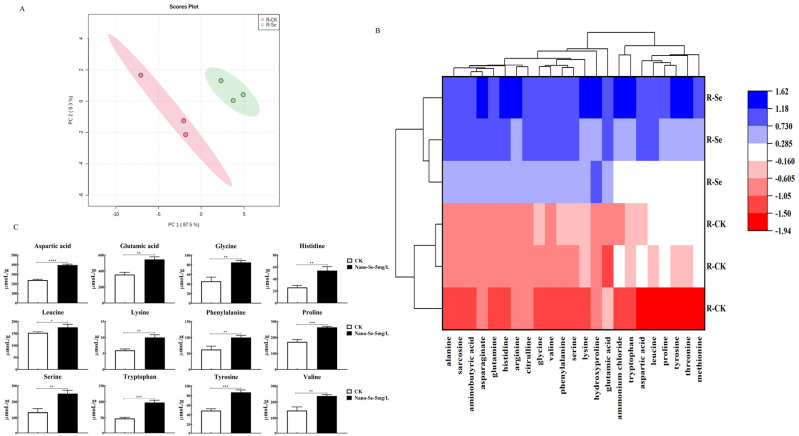
The PCA (**A**) and heatmap (**B**) displaying changes in amino acids. Nano-Se (5 mg/L) treatment notably enhanced levels of 12 amino acids (**C**). Data are presented as mean ± SEM (* *p* < 0.05, ** *p* < 0.01, *** *p* < 0.001, **** *p* < 0.0001. *n* = 3).

**Figure 5 foods-11-03019-f005:**
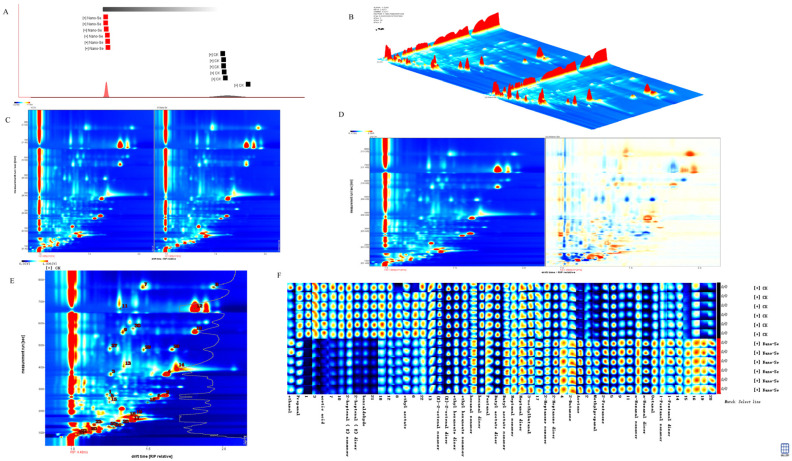
The PCA analysis (**A**), GC-IMS three-dimensional spectrum (**B**), vertical view (**C**), gas phase ion migration spectroscopy (**D**), qualitative result diagram of gas phase ion mobility spectrum (**E**), and gallery plot (**F**) by the nano-Se treatment of *S. grosvenorii*. Data are presented as mean ± SEM (*n* = 3).

**Figure 6 foods-11-03019-f006:**
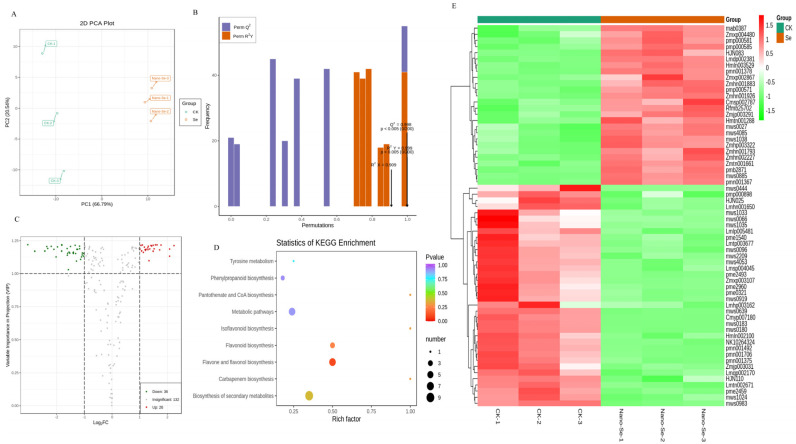
The principal component analysis chart of two treatments (**A**), OPLS-DA model validation diagram (**B**), volcanic maps of different metabolites (**C**), and the KEGG enrichment map of different metabolites (**D**) in *S. grosvenorii*. Cluster heat maps of different compounds (**E**) showed the change of metabolic compounds. Data are presented as mean ± SEM (*n* = 3).

## Data Availability

The data that support the findings of this study are included in the paper.

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
