# Peer review of "Metabolomic Analysis on the Mechanism of Nanoselenium Biofortification Improving the Siraitia grosvenorii Nutritional and Health Value"

_foods, 2022, doi:10.3390/foods11193019_

Round 1

Reviewer 1 Report

The manuscript has been very well prepared, it is of high quality. The only suggestion I can make is about figures.

Line 481: Figure 1 - authors need to change the font size "Figure 1" is way too large.

I'm also a little bit concerned about the readability of the diagrams in Figure 4 A-B (line 515) as the axis descriptions are bearly visible. The same is Figure 5 A-F (line 524) and Figure 6 A-E (line 526).

Reviewer 2 Report

The manuscript is about the effect of nano-selenium (Nano-Se) foliar application in the quality of nutrients, volatile organic compounds and how the foliar application of nano-se adjusting phenylpropane. The experiment showed the application of Nano Selenium (Nano-S ) has a beneficial effect in of Siraitia grosvenorii nutritional and health compounds.

1. The experiment has only two treatments Control and nano-Se 0.5 mg/L.

2.The authors must use T-student for statistical analysis,because the experiment has only two treatment the control and nano-Se in one concentration.

3.The conclusion the lines 345-346. The authors should erase:This study...... S. grosvenorii

Reviewer 3 Report

This is a good manuscript. The results demonstrated that the nano-Se has significant enhancement on carbohydrates, amino acids, and aromatic substances. The reviewer suggests that the authors should describe the the data in more details and present some key data in a tablet format. As can be seen, most of the work is a comparison with and without nano-Se (5 mg/L). Is there the effect is expected at various concentration? Although the main focus of this work was not reveal the enhancing mechanism, any discussion on the mechanism would have a big impact.

Round 2

Reviewer 2 Report

The corrections were done